# Warm/cool-tone switchable thermochromic material for smart windows by orthogonally integrating properties of pillar[6]arene and ferrocene

Sai Wang[1], Zuqiang Xu[2], Tingting Wang[2], Tangxin Xiao[3], Xiao-Yu Hu[2], Ying-Zhong Shen[1] & Leyong Wang[2,3]

Functional materials play a vital role in the fabrication of smart windows, which can provide a more comfortable indoor environment for humans to enjoy a better lifestyle. Traditional materials for smart windows tend to possess only a single functionality with the purpose of regulating the input of solar energy. However, different color tones also have great influences on human emotions. Herein, a strategy for orthogonal integration of different properties is proposed, namely the thermo-responsiveness of ethylene glycol-modified pillar[6]arene (**EGP6**) and the redox-induced reversible color switching of ferrocene/ferrocenium groups are orthogonally integrated into one system. This gives rise to a material with cooperative and non-interfering dual functions, featuring both thermochromism and warm/cool tone-switchability. Consequently, the obtained bifunctional material for fabricating smart windows can not only regulate the input of solar energy but also can provide a more comfortable color tone to improve the feelings and emotions of people in indoor environments.

[1] Applied Chemistry Department, School of Material Science & Engineering, Nanjing University of Aeronautics & Astronautics, Nanjing 210016, China. [2] Key Laboratory of Mesoscopic Chemistry of MOE, Collaborative Innovation Center of Chemistry for Life Sciences, School of Chemistry and Chemical Engineering, Nanjing University, Nanjing 210023, China. [3] School of Petrochemical Engineering, Changzhou University, Changzhou 213164, China. Correspondence and requests for materials should be addressed to X.-Y.H. (email: huxy@nju.edu.cn) or to Y.-Z.S. (email: yz_shen@nuaa.edu.cn) or to L.W. (email: lywang@nju.edu.cn)

S mart windows have great potential as a way of improving building energy efficiency and indoor comfort, by reversibly modulating the amount of visible light and solar radiation entering into the buildings[1–5]. To achieve switchable optical performance of such smart windows, avoiding unnecessary energy consumption and unsatisfactory indoor environment, a large amount of prior research has been performed, which mostly focused on controlling the transmittance of windows by external stimuli, including electricity (electrochromism)[6–10], light (photochromism)[11–13], heat (thermochromism)[14–16], and gas (gasochromism)[17]. Amongst these, thermochromic smart windows have attracted specific attention because they can automatically adjust the window transparency through weather-responsive control without extra energy input. This results in a reversible regulation of the solar transmittance (mainly in the near-infrared region), automatically adjusting the amount of solar heat entering into the buildings[18–20]. So far, vanadium dioxide ($VO_2$) is one of the most widely used inorganic materials for the fabrication of thermochromic smart windows, but the application of $VO_2$-based smart windows is limited by many shortcomings, such as a high transition temperature (68 °C), low light transmittance, and unattractive blackness[21–23]. Meanwhile, to the best of our knowledge, few studies have achieved the construction of smart windows based on organic materials, and most of the known systems are limited to a regulation of solar energy input, thus only possessing a single functionality[24–30].

However, taking into account emotional and psychological effects, not only the amount of light input, but also the color of the light has a tremendous importance. In color science, color can be categorized into warm color, cool color, and neutral color[31,32]. Warm-tone colors, such as red, orange, and yellow makes people feel warm, lively, and excited, while cool-tone colors including green, blue, and purple gives us a feeling of coolness, serenity, and calmness. Smart windows that are capable of providing warm colors in winter and cool colors in summer will not only make our homes much more comfortable, but can also be beneficial to improve human emotions. Although, thermochromic supramolecular materials based on charge-transfer interactions with excellent color switchable property developed by Fang and Olson have attracted great interest and attention[33,34], their application in the fabrication of smart windows has yet to be realized. Accordingly, developing an ideal warm/cool tone switchable thermochromic material for fabricating smart windows is an amazing and challenging endeavour, which could have an immediate positive effect on improving the quality of life for people throughout the world.

Orthogonal self-assembly refers to an integration strategy of two or more types of non-covalent interactions with high specificity and selectivity[35–42]. Using this orthogonal strategy, various types of supramolecular complexes (such as ligand-receptor, host–guest) have been applied for the fabrication of smart and functional materials based on supramolecular polymers. However, up to now, most orthogonal self-assembly strategies rely on selective orthogonal supramolecular interactions, while the individual functional properties of the interacting entities are not fully explored. For example, pillar[n]arenes[43–53], as a family of macrocyclic hosts, not only possess outstanding properties in host–guest chemistry, but can also undergo facile modification to introduce additional fascinating functionalities, such as hydrophobicity for super-hydrophobic surfaces[54], water-solubility[55], biofilm inhibition[47], multi-responsiveness to $CO_2$[56], light[48], temperature[57], and so on. Inspired by the concept of orthogonal self-assembly, we try to orthogonally integrate the unique properties of different supramolecular substrates to achieve the fabrication of smart materials with cooperative and non-interfering functions.

Herein, a dry gel with ferrocene pendants modifying on the polyacrylamide-based polymer networks (**Fc-gel**) has been successfully prepared. A well-swollen and high transparent hydrogel (**Fc-gel·EGP6**) is obtained when the **Fc-gel** is immersed in the

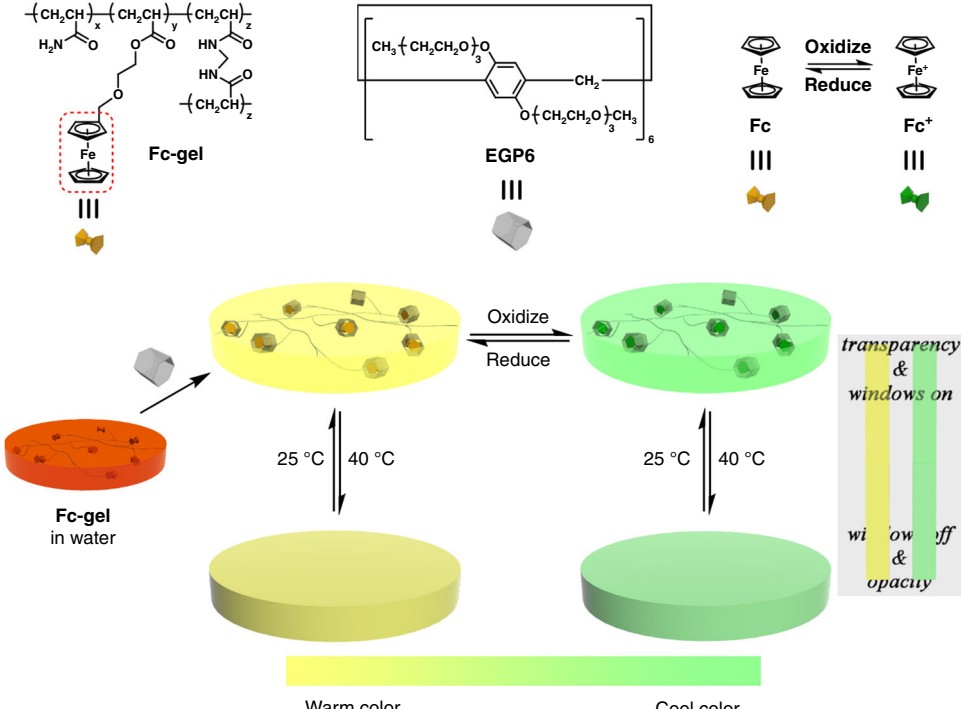

**Fig. 1** Chemical structures and schematic illustration of the warm/cool tone switchable thermochromic material for smart windows. Preparation of **Fc-gel•EGP6** hydrogel and the schematic illustration of warm/cool tone switchable thermochromic material based on the orthogonal integration strategy of the unique properties of pillar[6]arene and ferrocene

aqueous solution of ethylene glycol chain-modified pillar[6]arene (**EGP6**). This phenomenon is due to the formation of hydrophilic inclusion complexes between **EGP6** and the ferrocene groups of **Fc-gel** instead of the original hydrophobic domains formed by physical cross-linking of the hydrophobic ferrocene units. Thus, water absorption capacity of the generated **Fc-gel·EGP6** hydrogel is significantly improved, which results in a drastically improved swelling behaviour and an enhanced transparency of the **Fc-gel·EGP6** hydrogel in comparison to the dry **Fc-gel**. Moreover, owing to the thermo-responsiveness of **EGP6** and the reversible transformation between orange and green colors of ferrocene/ferrocenium groups under redox-response, we have orthogonally integrated the functional properties of both supramolecular building blocks based on host–guest interaction in order to design a warm/cool tone-switchable thermochromic material. This material is highly suitable for the fabrication of smart windows with dual functionality, both for regulating the input of solar energy and for improving the feelings and emotions of inhabitants for a comfortable life (Fig. 1).

## Results

**Preparing Fc-gel·EGP6 hydrogel by host–guest complexation.** Initially, the complexation between **EGP6** and **Fc**-containing polymer networks was investigated by $^1H$ NMR spectroscopy. Since both the polymer networks and the hydrophobic ferrocene moiety are insoluble in water, a water-soluble linear polymer **mPEG-Fc** was selected as a model guest for the host–guest complexation study (Fig. 2 and Supplementary Fig. 2). When a mixture of **EGP6** and **mPEG-Fc** with a 1:1 mole ratio was dissolved in $D_2O$, the peaks for protons on **EGP6** were shifted to low field, with the most significant downfield chemical shift observed for the aromatic protons $H_A$ ($\Delta\delta = 0.14$ ppm). Meanwhile, the signal for the ferrocene-protons ($H_B$, including one methylene unit) of **mPEG-Fc** showed an obvious upfield chemical shift (overlapped with the peaks of **EGP6**) due to the shielding effect of the electron-rich cavity of **EGP6**. The above results reveal that the ferrocene moiety of model guest **mPEG-Fc** is threaded into the

hydrophobic cavity of **EGP6**. This was also confirmed by a 2D NOESY experiment (Supplementary Figs. 4 and 5), which clearly showed the NOE correlation signals between protons $H_A$ of **EGP6** and part of protons $H_B$ on the ferrocene moiety of **mPEG-Fc**.

Subsequently, a Job's plot analysis based on UV–vis spectra data showed that the complexation between **EGP6** and the ferrocene moiety of **mPEG-Fc** had 1:1 stoichiometry (Supplementary Fig. 1). The corresponding binding constant of **EGP6** ⊃ **mPEG-Fc** in water was determined to be $(3.98 \pm 0.10) \times 10^2\ M^{-1}$ by a non-linear curve-fitting method (Supplementary Fig. 3). In this case, hydrophobic interaction is the main driving forces for such a host–guest complexation, leading to the formation of a stable 1:1 inclusion complex between **EGP6** and the ferrocene moiety.

Having demonstrated the **Fc-EGP6** interaction in the model substrate, we turned our attention to the **Fc**-substituted polymer networks for the following investigation. A disc-shaped sample consisting of cross-linked random copolymers (**Fc-gel**) was prepared by free radical copolymerization of acrylamide, a ferrocene-modified monomer, and $N,N'$-methylenebis(acrylamide) (MBA) under conventional conditions (for details, see preparation of **Fc-gel** in Supplementary Discussion). Based on our previous work[55], the obtained **Fc-gel** should display a significant swelling behavior in an aqueous solution of water-soluble pillar[6]arene. The swelling ratio will be dramatically promoted with an increasing amount of the incorporated ferrocene subunits from 0% to 10 mmol%; whereas, when the ferrocene loading was more than 10 mmol%, a decreased swelling ratio is expected induced by the dominant hydrophobic effect of the ferrocene groups. Therefore, in this study, the molar ratio of ferrocene subunits in **Fc-gel** was selected as 10 mmol% for sufficient host–guest complexation with **EGP6** and subsequent excellent swelling behavior of **Fc-gel**.

The samples of **Fc-gel** were immersed in water or **EGP6** aqueous solution at 25 °C to investigate their different swelling behaviors. When immersed in **EGP6** aqueous solution (15 mM),

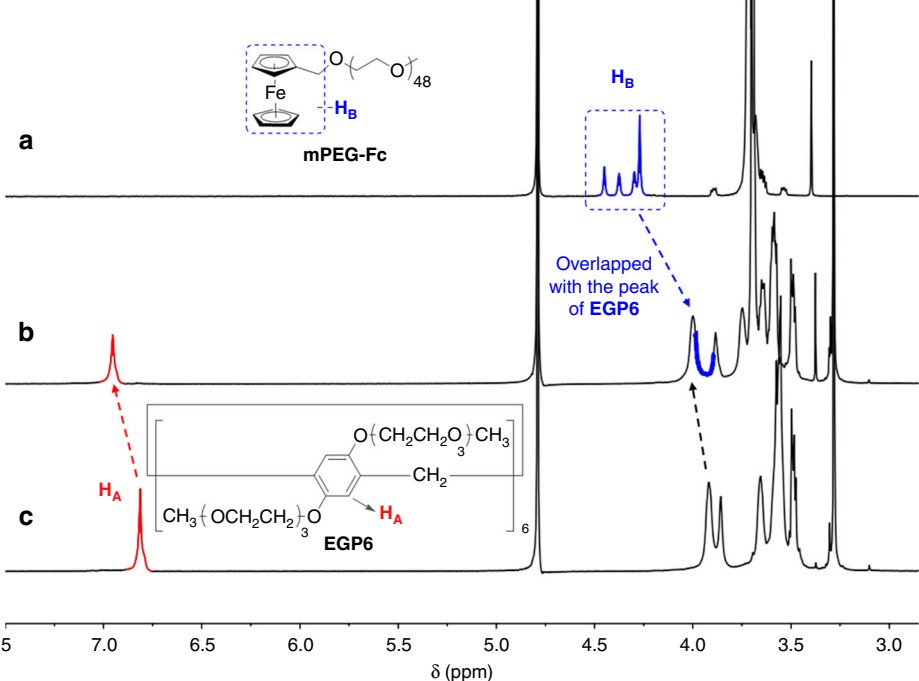

**Fig. 2** Host–guest complexation investigated by $^1H$ NMR spectroscopy. Partial $^1H$ NMR spectra (400 MHz, $D_2O$, 298 K) of **a** 2 mM **mPEG-Fc**; **b** 2 mM **EGP6** and 2 mM **mPEG-Fc**; **c** 2 mM **EGP6**

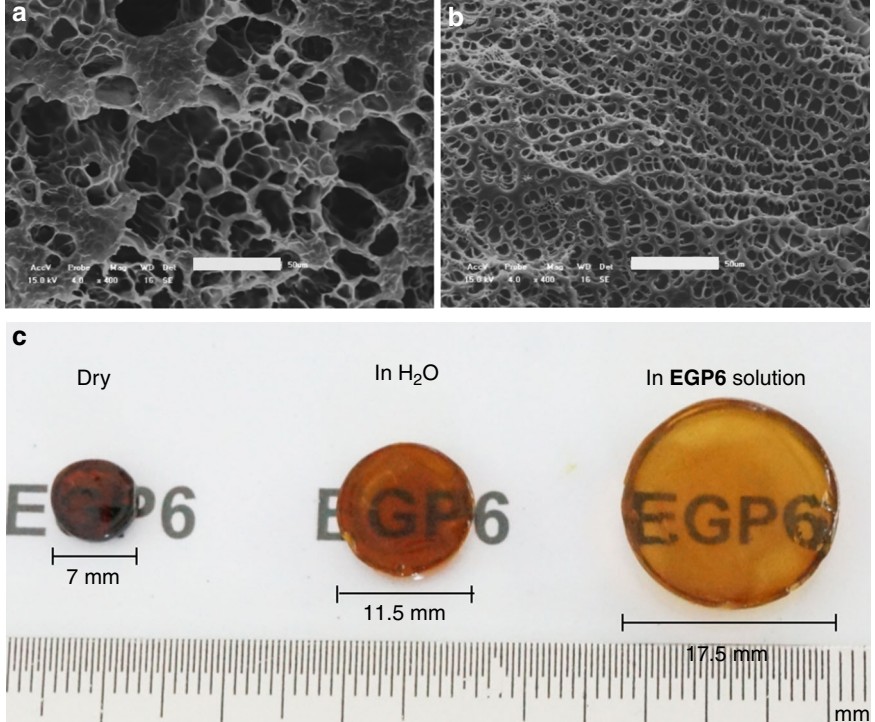

**Fig. 3** SEM micrographs and photographs of hydrogels. SEM micrographs of freeze-dried **Fc-gel** immersed in **a** pure water and **b** EGP6 aqueous solution (15 mM); Scale bars are 50 μm. **c** the corresponding photographs of **Fc-gel** under different conditions

the dry **Fc-gel** showed dramatical swelling, and the diameter of the disc increased from 7 to 17.5 mm with an increment of 150% compared to the dried sample (Fig. 3c and Supplementary Fig. 6). Moreover, the obtained sufficiently swollen hydrogel exhibited a homogeneous morphology with a porous network structure, as observed from scanning electron microscopy (SEM) image (Fig. 3b). In contrast, when the dry **Fc-gel** was immersed in pure water as a control experiment, the equilibrium diameter only increased by 64.3% (from 7 to 11.5 mm, Fig. 3c and Supplementary Fig. 6), and the obtained hydrogel exhibited a heterogeneous morphology with denser matrix and a non-uniform porous structures observed from the SEM image (Fig. 3a). The main reason for this different swelling behavior can be attributed to the formation of hydrophilic **EGP6**-ferrocene inclusion complexes, which transformed the hydrophobic ferrocene groups into hydrophilic moieties and resulted in the obviously improved water absorption capacity and drastically increased swelling property of the **Fc-gel** in the presence of **EGP6**.

**Fabricating orthogonal functional material for smart windows.** The response to external stimuli is a significant property of smart hydrogels. Since **EGP6** shows a unique thermo-responsiveness[57] and the ferrocene moiety exhibits a reversible transformation between orange and green ferrocene/ferrocenium groups under redox-stimuli[55], we envisaged the orthogonal integration of both of the above unique properties to design a warm/cool tone-switchable thermochromic material for fabricating smart windows. Initially, we investigated the stimuli-responsive properties of **EGP6** and ferrocene in aqueous solution. When **EGP6** aqueous solution was heated up to 40 °C, a white and turbid solution was obtained, which could return to a clear solution after being cooled to 25 °C (Fig. 4a, b). Meanwhile, **mPEG-Fc** in its aqueous solution exhibited an orange color. Whereas, upon oxidization with ammonium persulfate, the original orange solution gradually became dark green due to the transformation of ferrocene into ferrocenium ($Fc^+$), which could be

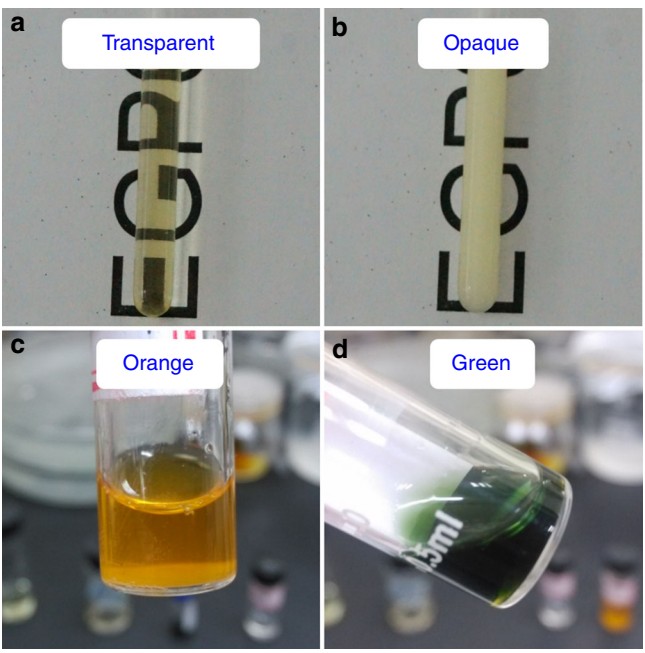

**Fig. 4** Stimuli-responsive properties of EGP6 and ferrocene in solution. Transparency of **EGP6** aqueous solution (4 mM): **a** at 25 °C and **b** at 40 °C; colors of **mPEG-Fc** solution at **c** reduction state and **d** oxidation state, respectively

reversed by subsequent reduction process, restoring the original orange color (Fig. 4c, d, for the host–guest interaction between $Fc^+$ and **EGP6**, see Supplementary Figs. 8–10). Based on these observations, we tried to orthogonally integrate the above unique properties of **EGP6** and ferrocene to the obtained hydrogel for further fabricating of a warm/cool tone-switchable thermochromic material for smart windows.

For the construction of smart windows, transparency is one of the most important prerequisites to ensure indoor illumination and input of solar energy. Therefore, the prepared functional materials for smart windows should firstly be transparent. The obtained well-swollen **Fc-gel·EGP6** hydrogel shows a high grade of transparency and an orange color, thus providing an excellent foundation for its potential application in constructing windows (Fig. 3c). Investigation of the optical transparency of **Fc-gel·EGP6** hydrogel by UV–vis transmission spectrum (Supplementary Fig. 7) showed enhanced transmittance with increasing wavelength in the visible region (380–780 nm) at 25 °C. More importantly, the hydrogel showed a high transmittance from 79.4 to 89.6% in the near infrared region (780–1100 nm). Thus, the **Fc-gel·EGP6** hydrogel is ideally suited to allow an excellent input of solar energy, effectively increase the indoor temperature with windows prepared by such a functional material. Moreover, the orange warm-toned color of the material makes it a potential candidate for smart windows with the ability to make indoor people feel warm and lively.

We next turned our attention to a possible thermo-responsiveness of the material. From an application point-of-view, we aspired to minimize the input of solar energy when the indoor temperature was too high to live comfortably. Especially in summer, thermo-responsive windows would represent an effective means of reducing the indoor temperature and thus decrease the electrical energy consumption of air conditioning devices. Therefore, we further evaluated the temperature-dependent transparency of **Fc-gel·EGP6** hydrogel caused by the thermal-responsiveness of **EGP6**. Upon increasing the environmental temperature to 40 °C, the **Fc-gel·EGP6** hydrogel rapidly became opaque within only 10 s (Fig. 5b and Supplementary Movie 1), and the transmittance of the hydrogel approached zero in the whole wavelength range (300–2000 nm) (Fig. 6b and Supplementary Fig. 13). Notably, this hydrogel material showed a good combination of higher luminous transmittance ($T_{lum} = 64.0\%$) and solar modulation ability ($\Delta T_{sol} = 66.9\%$)[58], which was calculated according to the reference method[59]. However, in the control experiment, the **Fc-gel** hydrogel prepared by immersing in pure water always remained its transparent in a temperature range from 25 to 45 °C without any changes in transmittance

(Figs. 5a and 6a). This phenomenon is based on the thermal-responsiveness of **EGP6**. At low temperatures, the **Fc-gel·EGP6** hydrogel will be formed driven by host–guest interaction. However, upon increasing the temperatures above its cloud point temperature ($T_{cloud}$), **EGP6** will aggregate and microseparate from water inside the hydrogel due to the interaction of the hydrophobic groups of pillararene backbone[57], and finally, resulting in the generation of an opaque **Fc-gel·EGP6** hydrogel with the ability to prevent visible light and solar radiation from passing through. When decreasing the temperature from 40 to 25 °C, the opaque **Fc-gel·EGP6** hydrogel immediately returned to its original transparency because of the re-dissolution of **EGP6** below its $T_{cloud}$ (Supplementary Movie 1). Meanwhile, such reversible thermo-responsive behavior was also supported by variable-temperature $^1$H-NMR spectroscopy (Supplementary Fig. 11). More importantly, after more than 100 cycles of alternating temperature increasing/decreasing, the transmittances kept almost constant between two values measured at 1099 nm (Fig. 7a). In addition, the **Fc-gel·EGP6** hydrogel remained very good stability without any leakage of **EGP6** from the hydrogel backbone even after 100 thermal cycles (Supplementary Fig. 12). Therefore, we can confirm that the transparency/opaqueness of the **Fc-gel·EGP6** hydrogel is reversible and highly reproducible for many alternating cycles between 25 and 40 °C.

With regard to the second functionality of the material, the warm/cool tone color change, we subsequently investigated its reversible redox-behaviour. When the **Fc-gel·EGP6** hydrogel was immersed in ammonium persulphate aqueous solution ($(NH_4)_2S_2O_8$), the gel gradually changed its color from orange to green within 8 min (Fig. 5c and Supplementary Movie 2). This is mainly due to the fact that the original color of the hydrogel was contributed by the orange ferrocene group, which is transformed into green ferrocenium moiety upon oxidation (Fig. 4c, d). To our delight, the oxidation process had hardly any impact on the transmittance of the **Fc-gel·EGP6** hydrogel and the obtained oxidized hydrogel with green color also exhibited high transparency. From the UV–vis transmission spectra (Fig. 6c), a small absorption peak at about 620 nm could be observed, giving direct spectroscopic evidence of the ferrocenium-moiety. Meanwhile, similar as the **Fc-gel·EGP6** hydrogel, the oxidized hydrogel

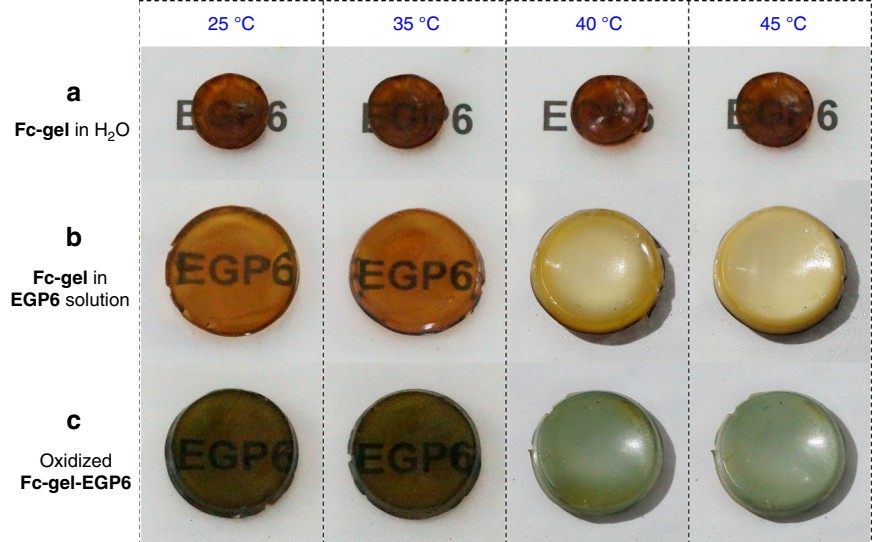

**Fig. 5** Warm/cool tone-switchable and thermochromic properties of the hydrogels. Transparency of **a Fc-gel** immersed in pure water (sample thickness 1 mm); **b Fc-gel** immersed in **EGP6** solution (forming **Fc-gel·EGP6** hydrogel, sample thickness 2.8 mm); **c** oxidized **Fc-gel·EGP6** hydrogel at 25 °C, 35 °C, 40 °C, and 45 °C, respectively

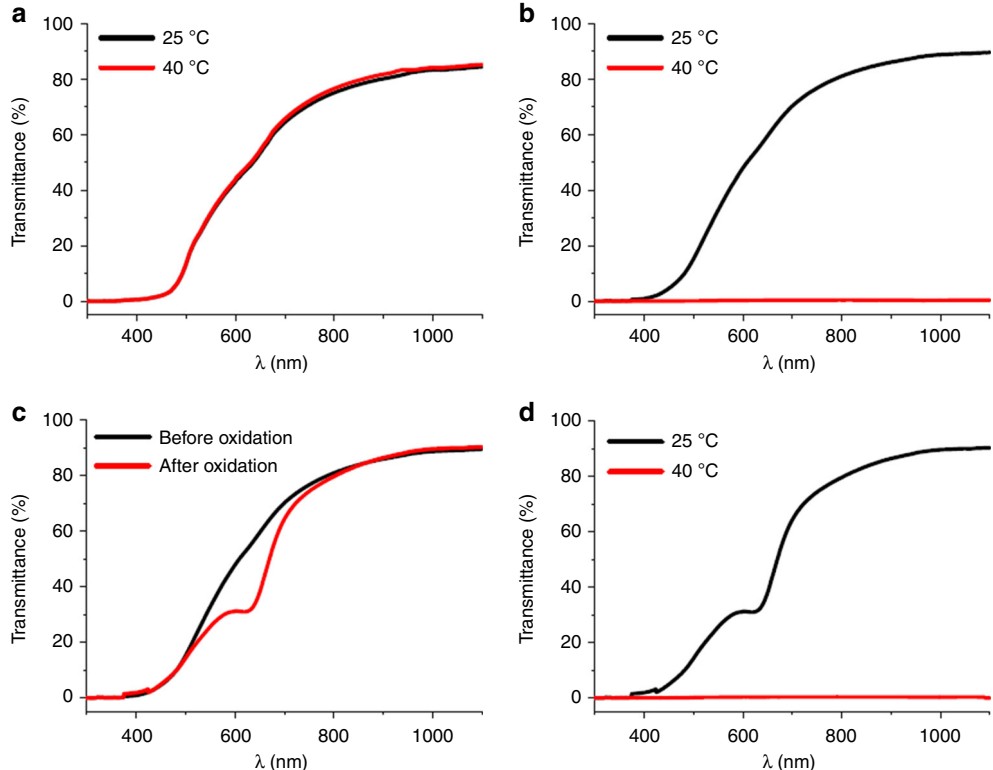

**Fig. 6** Transmittance changes under different conditions. Transmittance changes of **a** Fc-gel immersed in pure water; **b** Fc-gel•EGP6 hydrogel; **d** oxidized **Fc-gel•EGP6** hydrogel at 25 and 40 °C; **c** Fc-gel·EGP6 hydrogel before and after oxidation

with dark green color also had excellent thermochromism (Figs. 5c and 6d, and Supplementary Movie 3), and the reversible transformation between transparency and opaqueness of the oxidized hydrogel were also highly reproducible for more than 100 thermal cycles (Fig. 7b). Interestingly, a reverse process for transforming the green hydrogel into its original orange color could be achieved by immersing the oxidized hydrogel in hydrazine hydrate ($N_2H_4\cdot H_2O$) aqueous solution (Supplementary Movie 2), leading to the reduction of green ferrocenium into orange ferrocene groups[55]. And the regenerated orange hydrogel also maintained its high transparency and good performance of thermochromism (Fig. 7d). Moreover, the above switchable process between orange and green colors exhibited good reversibility and high repeatability based on the oxidization/ reduction of the ferrocene groups within the **Fc-gel·EGP6** hydrogel (Fig. 7c). More importantly, the oxidized green **Fc-gel·EGP6** hydrogel material possesses the characteristics of cool-toned colors. Thus, this material could be used to fabricate smart windows for providing cool and calm feelings of indoor people.

In addition, for the real-life applications of smart windows, a feasible and practical way for affecting the desired switching process is necessary. In this **Fc-gel·EGP6** hydrogel system, thermochromic performance depends on the environmental temperature, while the color switching between warm and cool tune can be controlled by a feasible redox stimuli. According to previous work[60–62], cyclic voltammetry (CV) study indicated that the pillararene–ferrocene inclusion complexes have stable oxidation and reduction peaks. Therefore, we have fabricated an electrochemical device based on the **Fc-gel•EGP6** hydrogel to achieve the color switching between warm and cool tune by applying a suitable electric potential instead of using chemical oxidizing and reducing agents. As shown in Fig. 8, after being immersed in KCl aqueous solution (0.1 M), the **Fc-gel·EGP6**

hydrogel was sandwiched between two pieces of ITO conductive glass to form a simple device. Upon holding the potential at +4.0 V for 6 min, the color of the material in the device could be completely transformed from orange to green due to the electrochemical oxidation of ferrocene to ferricenium cation. After further holding the potential at –4.0 V for 6 min, the **Fc-gel·EGP6** hydrogel was reduced and the color of the material returned to orange. Therefore, based on the orthogonally integrated properties of **EGP6** and ferrocene by host–guest interaction, a warm/cool tone-switchable **Fc-gel·EGP6** hydrogel was successfully prepared, which could be used as a novel material for fabricating smart windows to improve the emotions of indoor people, meanwhile, both in warm and cool tones, the hydrogel was transparent and possessed excellent thermochromism to control the input of solar energy for a more comfortable indoor environment.

## Discussion

In summary, we have orthogonally integrated the unique properties of pillar[6]arene and ferrocene based on host–guest interaction to develop a warm/cool tone-switchable thermochromic material for the fabrication of smart windows. Owing to the efficient **EGP6**-ferrocene complexation, a well-swollen and highly transparent **Fc-gel·EGP6** hydrogel was prepared by immersing a ferrocene-modified polyacrylamide-based polymer networks in **EGP6** aqueous solution. Moreover, the generated **Fc-gel·EGP6** hydrogel exhibited warm/cool tone-switchable property owing to the reversible transformation between orange and green ferrocene/ferrocenium groups under redox-stimuli, which has potential applications in fabricating smart windows with specific function for improving the feelings and emotions of indoor people. More importantly, the transparent **Fc-gel·EGP6** hydrogel inherited the excellent thermo-responsiveness of **EGP6** in both

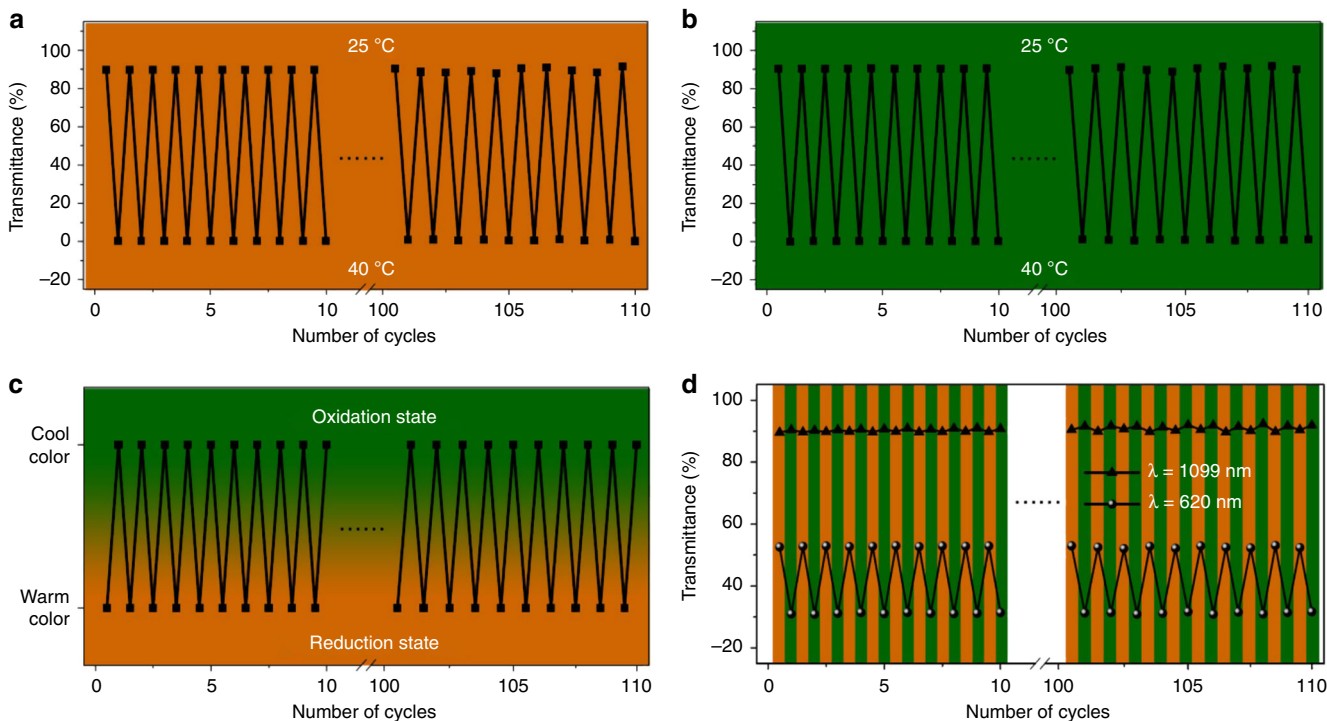

**Fig. 7** Reproducibility of the **Fc-gel·EGP6** hydrogel in warm/cool tone-switchable and thermochromic functions. Transmittance changes of **a** **Fc-gel·EGP6** hydrogel and **b** oxidized **Fc-gel·EGP6** hydrogel at 1099 nm by alternately exposing to water at 25 and 40 °C. **c** Color changes and **d** transmittance changes at 1099 and 620 nm of **Fc-gel·EGP6** hydrogel by alternately exposing to $(NH_4)_2S_2O_8$ (cool color) and $N_2H_4·H_2O$ (warm color) aqueous solution at 25 °C

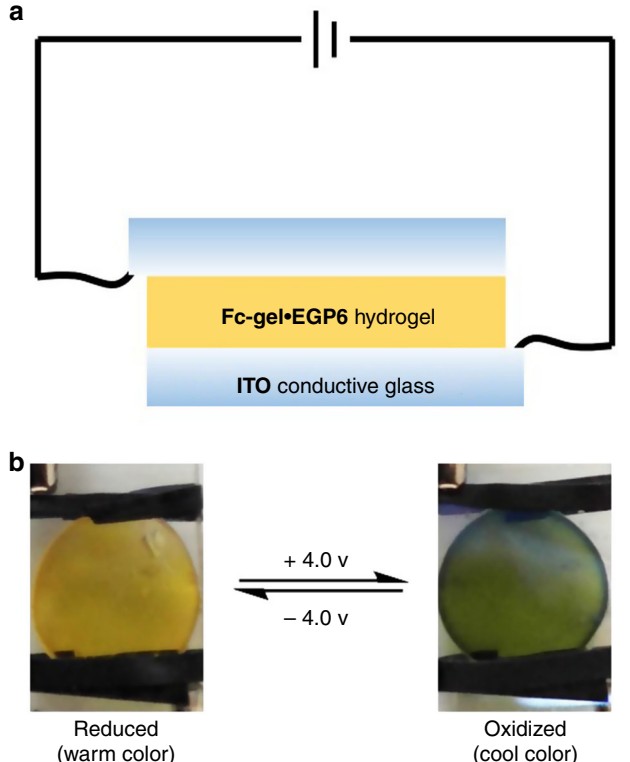

**Fig. 8** An electrochemical device based on **Fc-gel•EGP6** hydrogel. **a** Schematic illustration of the electrochemical device based on **Fc-gel•EGP6** hydrogel; **b** Photographs of the device in reduced and oxidized states, respectively

warm and cool color to become a thermochromic material, which can regulate the input of solar energy for a more comfortable indoor environment. In this study, the unique properties of **EGP6** and ferrocene were orthogonally integrated together, where both **EGP6** and ferrocene perform its own duty with non-interference but mutual promotion. Therefore, we successfully provide the first applied example for this orthogonal integration strategy of different properties with the purpose of developing a warm/cool tone-switchable thermochromic material for fabricating smart windows, which have promising applications in the field of material science and technology. Furthermore, we believe that this orthogonal integration strategy of different properties might provide a new direction for design and development of novel functional materials.

## Methods

**General**. All reagents were commercially available and used as supplied without further purification unless otherwise stated. **EGP6**, **mPEG-Fc** ($M_n = 2347$ g·mol$^{-1}$), ferrocene-modified acrylamide monomer (**FcAm**), and dry **Fc-gel** were synthesized according to the literatures[52,55,57]. Nuclear magnetic resonance (NMR) spectra were recorded on a Bruker Advance DMX 400 spectrophotometer or a Bruker Advance DMX 500 spectrophotometer with internal standard tetramethylsilane (TMS) and solvent signals as internal references at 25 °C. SEM was carried out on a Shimadzu SSX-550 device. UV–vis spectra were recorded on a Shimadzu UV-1780 UV–vis Spectrophotometer.

**Preparation method of dry Fc-gel**. FcAm (220 mg, 0.70 mmol), acrylamide (448 mg, 6.30 mmol), and N, N′-methylenebis(acrylamide) (MBA) (5.40 mg, 0.035 mmol) were dissolved in dimethyl sulfoxide (DMSO) (1.6 mL). Then azodiisobutyronitrile (AIBN) (28.8 mg, 0.175 mmol) was added to the solution and the mixture was purged with dry argon for 30 min. The resulting solution was equally divided into 10 vials (10 mm in diameter) and sealed. The polymerization was performed in an oven at 70 °C for 24 h. Then, the vials were cooled down to room temperature, disc-shaped samples were washed successively by DMSO and deionized water, and subsequently dried in oven (50 °C) under vacuum for 12 h after

natural drying. Finally, disc-shaped dry **Fc-gel** was obtained with 7 mm in diameter and 1 mm in thickness.

**Synthesis and characterization**. Synthesis and relevant characterization details are provided in the Supplementary Information.

**Data availability**. The authors declare that the data supporting the findings of this study are available within the paper and its Supplementary Information including Supplementary Methods and Supplementary Discussion. All data are available from the authors on reasonable request.

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

## Acknowledgements

This work was supported by the National Basic Research Program of China (2014CB846004) and the National Natural Science Foundation of China (No. 21472089 and 21572101). X-Y.H. also thanks the Alexander von Humboldt Foundation for a research fellowship. We would like to thank Dr. Jochen Niemeyer form University of Duisburg-Essen for his very kind help on English language polishing.

## Author contributions

L.W., X.-Y.H., and Y.-Z.S. conceived the project and supervised the research; S.W. designed and conducted the experiments; Z.X. performed parts of the synthetic experiments; T.W. performed the transmittance measurements; T.X. performed the drawing of the schematic diagram of hydrogels; S.W., X.-Y.H., and L.W. wrote the manuscript. All authors analyzed the data, discussed the results, and commented on the manuscript.

## Additional information

**Competing interests:** The authors declare no competing interests.

