## [Peer Review File · Nature Communications]

Reviewers' comments:

Reviewer #1 (Remarks to the Author):

In this very nice manuscript, the thermo-responsiveness of ethylene glycol chain-modified pillar[6]arene (EGP6) and the redox-induced reversible color switching between warm and cool tone of ferrocene/ferrocenium groups are orthogonally integrated into a system to achieve the fabrication of warm/cool tone-switchable thermochromic materials with cooperative and non-interfering dual functions. The obtained bifunctional material for fabricating smart windows can not only regulate the input of solar energy but also can provide a more comfortable in-door environment to improve the feelings and emotions of indoor people. Overall, the results presented in this manuscript are of interest to the readers. Furthermore, they have good scientific value and novelty. Therefore, I suggest the acceptance for publication after the following minor points are addressed:

1. One of the following important paper about pillararenes was missed: Acc. Chem. Res. 2012, 45, 1294–1308.
2. In this manuscript, the authors use ethylene glycol chain-modified pillar[6]arene (EGP6) to fabricate the hydrogel Fc-gel•EGP6 and claimed that the water absorption capacity of the generated Fc-gel•EGP6 hydrogel was significantly improved which resulted in the drastically swelling property and enhanced transparency of Fc-gel•EGP6 hydrogel than that of the dry Fc-gel. But there is no reasonable explanation about this dramatic phenomenon. So, a detailed investigation of the water absorption capacity and the enhancement of transparency of the Fc-gel•EGP6 hydrogel should be provided.
3. Ethylene glycol chain-modified pillar[6]arene (EGP6) shows thermo-responsiveness and the ferrocene moiety exhibits reversible transformation between orange and green ferrocene/ferrocenium groups under redox-stimuli in the Fc-gel•EGP6 hydrogel. Furthermore, the author said that EGP6 was used in this system to obtain the reported hydrogel because the host-guest interaction property between EGP6 and ferrocene. However, from the observation of present situation, the host-guest interaction between EGP6 and ferrocene made no contribution for the fabrication of the hydrogel. So, the review just wonders what the role of the host-guest interaction between EGP6 and ferrocene for the fabrication of the hydrogel is in this system.
4. Variable-temperature ¹H-NMR study should be performed to investigate the thermo-responsiveness of the host-guest interaction between EGP6 and ferrocene.
5. The authors investigated the optical transparency of Fc-gel•EGP6 hydrogel by UV-vis transmission spectrum and the hydrogel showed a high transmittance from 79.4% to 89.6% in the near infrared region. As we known, the transparency is proportional to the concentration of the compound contained in the material. So, the transparency of Fc-gel•EGP6 hydrogel at different concentrations of the gelators should be investigated here.
6. To construct smart windows, the stability of the hydrogel is one of the most important performance. So, the investigation of the stability of the hydrogel under different temperatures (25 °C–45°C as mentioned by the authors in the manuscript) was strongly recommended.
7. The host-guest interaction between Fc⁺ and EGP6 was investigated in Supplementary Figures 9–11, but the data presented here was about the study of EGP6 and Cob + PF6⁻, please check!

Reviewer #2 (Remarks to the Author):

The authors reported a new type of coating material potentially for smart window applications, which can switch color and transparency in response to temperature and redox species. While the integration of two stimuli-responsive materials to create dual-functional materials looks interesting, this work lacks of enough significance and novelty to be published in Nature Communication.

First of all, the idea of integrating two functional materials to develop multi-responsive materials is

not new. For example, in a paper recently published in *Advanced Functional Materials* (*Adv. Funct. Mater.* 2017, 27, 1702784), the authors also combined ferrocene derivative with a thermoresponsive polymer to develop a thermal/redox dual-functional material for smart windows. Depending on the salt concentration, the material becomes opaque at a temperature ranging from 33-66 C, demonstrating a better temperature tunability than the material reported in this manuscript.

Second, for smart window applications, the change of color by adding/immersing into redox species is not quite feasible. Also, the color change time (8 minutes) seems too long for real life applications. If the material can be electrochromic and can change its color at a faster speed, the importance of this work would be largely enhanced.

Lastly, although the authors showed some reproducibility result, the number of cycles is not enough for applications like smart windows. Reproducibility experiments with more cycles are encouraged.

Given the concerns mentioned above, I could not recommend publication of this work on *Nature Communication* at the present form.

Reviewer #3 (Remarks to the Author):

Manuscript by Sai Wang et al., " Warm/cool tone switchable thermochromic material for smart windows by orthogonally integrating the unique properties of pillar[6]arene and ferrocene " presents a novel orthogonal integration strategy of different properties for smart windows. The thermo-responsiveness of ethylene glycol chain-modified pillar[6]arene (EGP6) and the redox-induced reversible color switching between warm and cool tone of ferrocene/ferrocenium groups are integrated into a system to achieve the fabrication of warm/cool tone-switchable thermochromic materials with cooperative and non-interfering dual functions.

However, some details are not sufficient for publishing in the present form and major revision are needed. The following questions should be addressed before publishing.

1. The UV-Vis-NMR measurement is not in the full spectrum of solar energy (to 2500nm). Energy modulation need to be considered from visible range to NIR range.
2. From the paper, researchers only mentioned the thickness of dry Fc-gel, can you provide the thickness of samples in Fig 4? The very low luminous transmission is a concern. Reducing the thickness should help to enhance the luminous transmission to be used in thermochromic materials.
3. Durability is always a problem for organic smart window, for the durability test, 10 cycles were not enough for both transmittance change and redox reaction, is it possible to test at least 50 cycles or its better to test until the gel will not change color and transmittance to understand the limitation of this material (Fig 6)
4. Since it is a new type of thermochromics material, please characterize it in a standard way, please calculate the T_{lum} and $detaT_{sol}$. and the calculation details is listed in the paper <https://doi.org/10.1016/j.jallcom.2017.10.045> and compare with other thermochromic materials as summarized in 10.1039/C6TC02694J.
5. The authors provided a new idea to integrate the thermochromic composite with the color switchable compound. However, the idea is lack of proofing-of-concept. The authors demonstrated the gels with different oxidation state by soaking into different precursors. However, the capability of such a color-switch approaching to the real-life application has not been investigated. I suggest that the authors could fabricate device to show a demo that based on this new material to proof the concept.
6. To continue from previous point, How to immerse the hydrogel film into the respective solution for redox reactions in real application? Whereas, since this is a research paper, these concerns are less critical. Maybe share a few strategies to put this material in real applications.

The following is a point-to-point response to the reviewers' comments (for your convenience, we repeat the referee's comments below in *black*, followed by our reply in *blue*).

Reviewer 1

Comments to the Author

“In this very nice manuscript, the thermo-responsiveness of ethylene glycol chain-modified pillar[6]arene (EGP6) and the redox-induced reversible color switching between warm and cool tune of ferrocene/ferrocenium groups are

orthogonally integrated into a system to achieve the fabrication of warm/cool tone-switchable thermochromic materials with cooperative and non-interfering dual functions. The obtained bifunctional material for fabricating smart windows can not only regulate the input of solar energy but also can provide a more comfortable in-door environment to improve the feelings and emotions of indoor people. Overall, the results presented in this manuscript are of interest to the readers. Furthermore, they have good scientific value and novelty. Therefore, I suggest the acceptance for publication after the following minor points are addressed.”

Response:

We greatly appreciate Reviewer 1 for his/her positive evaluation and kind recommendation of publishing our manuscript in *Nature Communications* after minor revisions. And all of the issues suggested by Reviewer 1 have been addressed accordingly in detail as follows, and the revisions have also been made in the Revised Manuscript and Supplementary Information.

General issues:

1. “One of the following important paper about pillararenes was missed: *Acc. Chem. Res.* 2012, 45, 1294–1308.”

Response:

Thanks a lot for the comments. As suggested, this important paper related to pillararenes (*Acc. Chem. Res.* 2012, 45, 1294-1308) has been cited as ref. 50 in the revised Manuscript.

2. “In this manuscript, the authors use ethylene glycol chain-modified pillar[6]arene (**EGP6**) to fabricate the hydrogel **Fc-gel•EGP6** and claimed that the water absorption capacity of the generated **Fc-gel•EGP6** hydrogel was significantly improved which resulted in the drastically swelling property and enhanced transparency of **Fc-gel•EGP6** hydrogel than that of the dry **Fc-gel**. But there is no reasonable explanation about this dramatic phenomenon. So, a detailed investigation of the water absorption capacity and the enhancement of transparency of the **Fc-gel•EGP6** hydrogel should be provided.”

Response:

Many thanks for the profound suggestions. According to reviewer 1’s suggestion, we have provided more detailed explanations about this dramatic phenomenon in the revised Manuscript. Upon adding **EGP6** to the **Fc-gel**, due to the formation of hydrophilic **EGP6**-ferrocene inclusion complexes, the original hydrophobic ferrocene groups were transformed into hydrophilic moieties, which resulted in the obviously improved water absorption capacity and drastically swelling property. On the other hand, the original hydrophobic domains of **Fc-gel** were formed by physical cross-linking of the hydrophobic ferrocene groups and the gel exhibited a heterogeneous structure as confirmed by SEM image, which resulted in the light scattering, light

refraction, and light reflection. However, with respect to the formed **Fc-gel•EGP6** hydrogel, the hydrophilic **EGP6**-ferrocene inclusion complexes eliminated the original hydrophobic domains and the whole polymer chains could be well integrated into water, so that the polymer chains had hardly any effects on the transmittance of the generated **Fc-gel•EGP6** hydrogel. Thus, the transparency of **Fc-gel•EGP6** hydrogel was enhanced compared to that of the dry **Fc-gel**. In addition, the homogeneous morphology with uniform porous network structure of the formed **Fc-gel•EGP6** hydrogel further confirmed the above point.

3. “Ethylene glycol chain-modified pillar[6]arene (**EGP6**) shows thermo-responsiveness and the ferrocene moiety exhibits reversible transformation between orange and green ferrocene/ferrocenium groups under redox-stimuli in the **Fc-gel•EGP6** hydrogel. Furthermore, the author said that **EGP6** was used in this system to obtain the reported hydrogel because the host–guest interaction property between **EGP6** and ferrocene. However, from the observation of present situation, the host–guest interaction between **EGP6** and ferrocene made no contribution for the fabrication of the hydrogel. So, the review just wonders what the role of the host–guest interaction between **EGP6** and ferrocene for the fabrication of the hydrogel is in this system.”

Response:

Thanks a lot for the profound comments. In this system, the host–guest interaction between **EGP6** and ferrocene was a necessary factor for the fabrication of the swollen hydrogel. Firstly, the host–guest interaction between **EGP6** and ferrocene moiety was confirmed in aqueous solution by using a water-soluble linear polymer **mPEG-Fc** as a model guest. And then, when dry **Fc-gel** was immersed in the **EGP6** aqueous solution, **EGP6** could interact with the ferrocene moiety in the **Fc-gel** to form hydrophilic **EGP6**-ferrocene inclusion complexes. Thus, the original hydrophobic domains formed by ferrocene groups disappeared, and accordingly, the **Fc-gel** could absorb a large amount of water and swelled dramatically to achieve the fabrication of **Fc-gel•EGP6** hydrogel with enhanced transparency. In addition, some similar examples based on host–guest interactions have also been reported in previous works (*J. Am. Chem. Soc.* **2016**, *138*, 6643–6649; *Angew. Chem. Int. Ed.* **2013**, *52*, 8961-8963).

4. “Variable-temperature ¹H-NMR study should be performed to investigate the thermo-responsiveness of the host–guest interaction between **EGP6** and ferrocene.”

Response:

Thanks a lot for the helpful suggestions. The thermo-responsiveness of the host–guest interaction between **EGP6** and ferrocene moiety were investigated by variable-temperature ¹H-NMR spectroscopy (**Figure R1**). The results showed that when an aqueous solution of **EGP6** and **mPEG-Fc** was heated up to 45 °C, the

chemical shifts of ferrocene signals of the model guest **mPEG-Fc** returned to the uncomplexed state. However, the complexation between **EGP6** and **mPEG-Fc** re-formed after decreasing the solution temperature to 25 °C. Therefore, the complexation between **EGP6** and **mPEG-Fc** can be reversibly controlled by heating and cooling. In addition, **Figure R1** and the corresponding descriptions were also added in the revised Supplementary Information as **Supplementary Figure 12**.

Figure R1. Partial variable temperature ^1H NMR spectra (600 MHz, D_2O) of (a) **mPEG-Fc** (4 mM) at 45 °C, and **mPEG-Fc** (4 mM) with the presence of **EGP6** (2 mM) at: (b) 45 °C; (c) 40 °C; (d) 35 °C; (e) 30 °C; (f) 25 °C.

5. “The authors investigated the optical transparency of **Fc-gel•EGP6** hydrogel by UV-vis transmission spectrum and the hydrogel showed a high transmittance from 79.4% to 89.6% in the near infrared region. As we known, the transparency is proportional to the concentration of the compound contained in the material. So, the transparency of **Fc-gel•EGP6** hydrogel at different concentrations of the gelators should be investigated here.”

Response:

Many thanks for the profound comments and suggestions. According to Reviewer 1’s suggestions, the transparency of **Fc-gel•EGP6** hydrogel with the presence of different concentrations of **EGP6** (0 mM, 5 mM, 10 mM, 15 mM, and 20 mM) was investigated by using UV-vis transmission spectrum as shown in Figure R2, and the results were also added in the revised Supplementary Information as Fig. 7 instead of the original Supplementary Fig. 7. For the **Fc-gel**, the original hydrophobic domains were formed by physical cross-linking of the hydrophobic ferrocene groups and the gel exhibited a heterogeneous structure as confirmed by

SEM image, which led to the light scattering, light refraction and reflection. When the **Fc-gel** was immersed in pure water, the polyacrylamide-based main chains could absorb water, which caused that the heterogeneous structure was improved and the transmittance of the hydrogel was enhanced. Upon adding **EGP6** solution to the **Fc-gel**, the original hydrophobic domains composed by ferrocene groups gradually transformed into hydrophilic **EGP6-ferrocene** inclusion complexes, which dramatically improved the water absorption ability of **Fc-gel** and the obtained well-swollen **Fc-gel•EGP6** hydrogel showed enhanced transparency. From Figure R2, it was found that the transmittance of hydrogel was enhanced more efficiently with a low concentration of **EGP6** than that of with a higher concentration. And at higher concentrations of 15 mM and 20 mM, there was hardly any enhancement in the transmittances. The main reason might be that the heterogeneous structure could be improved more efficiently at beginning upon adding **EGP6**, while the remained a small portion of heterogeneous structure was too less to be transformed for enhancing the transmittance of the hydrogel. In addition, the above investigations have been added in the revised Supplementary Information as Supplementary Fig. 7 in Part S4.

Figure R2. Transmittances of sufficiently swollen hydrogel by immersing in pure water or **EGP6** solution of different concentrations at 25 °C.

6. “To construct smart windows, the stability of the hydrogel is one of the most important performance. So, the investigation of the stability of the hydrogel under different temperatures (25 °C-45 °C as mentioned by the authors in the manuscript) was strongly recommended.”

Response:

Many thanks for the profound suggestions. The main factor determining the stability of **Fc-gel•EGP6** hydrogel is its capability to hold **EGP6** within the hydrogel without leaking out from the hydrogel backbone.

To investigate the stability of **Fc-gel•EGP6** hydrogel, we alternately exposed the hydrogel to air at 25 °C and then to water at 40 °C for 50 and 100 cycles, respectively. Then the water medium was measured by UV-Vis Spectroscopy. As shown in Figure R3, there was almost no absorption signal at 290 nm (a characteristic absorption peak of **EGP6**), which revealed that almost no **EGP6** had leaked out from the hydrogel. So the stability of the **Fc-gel•EGP6** hydrogel is very good. In addition, the above investigation of the stability of the hydrogel under different temperatures was also added in the revised Supplementary Information as Supplementary Fig. 13.

Figure R3. UV-Vis absorption spectra of **EGP6** solution and the water medium after different thermal cycles upon alternately exposing the **Fc-gel•EGP6** hydrogel to air at 25 °C and then to the water medium at 40 °C.

7. “The host–guest interaction between Fc^+ and **EGP6** was investigated in Supplementary Figures 9–11, but the data presented here was about the study of **EGP6** and $\text{Cob}^+\text{PF}_6^-$, please check!”

Response:

Many thanks for the helpful suggestions. According to Reviewer 1’s suggestion, we have checked the data in the original Supplementary Information (**Part S6**). There may be some misunderstanding, because we have stated in the Supplementary Information (page S8) as follows “Due to the paramagnetic property of Fc^+ , it is difficult to study the host–guest complexation between **EGP6** and Fc^+ . In this case, the diamagnetic cobaltocenium ion (Cob^+), which has a similar binding ability to pillararene as that of Fc^+ , (*Chem. Commun.* 2013, 49, 5085) was selected as an analogue of Fc^+ .” Therefore, the host–guest complexation between **EGP6** and Fc^+ was investigated by using cobaltocenium hexafluorophosphate ($\text{Cob}^+\text{PF}_6^-$) as an analogue of Fc^+ based on the ^1H NMR spectroscopy.

Referee 2

Comments to the Author

“The authors reported a new type of coating material potentially for smart window applications, which can switch color and transparency in response to temperature and redox spices. While the integration of two stimuli-responsive materials to create dual-functional materials looks interesting, this work lacks of enough significance and novelty to be published in Nature Communication.”

Response:

We greatly appreciate Reviewer 2’s comments. However, we couldn’t bring ourselves to agree with his/her view that *“this work lacks of enough significance and novelty”*. The idea proposed in our work is based on an orthogonal integration strategy of different properties, which was proposed for the first time. In this work, the thermo-responsiveness of **EGP6** and the redox-induced reversible color switching between warm and cool tone of ferrocene/ferrocenium units were orthogonally integrated into a supramolecular system to achieve the fabrication of warm/cool tone-switchable thermochromic materials with cooperative and non-interfering dual functions. The obtained hydrogel material could be potentially applied for fabricating smart windows which can not only regulate the input of solar energy but also can provide a more comfortable indoor environment to improve the feelings and emotions of indoor people. Meanwhile, both in warm and cool tones, temperature regulation could be achieved by switching the transmittance of **Fc-gel•EGP6** hydrogel between transparency and opacity. Notably, these two functions were cooperative and non-interfering. So we believe that this orthogonal integration strategy of different properties could provide a new direction for design and development of novel functional materials. In addition, according to the profound suggestions of Reviewer 2, we have further improved the quality of our manuscript and demonstrated the significance and novelty of this work in details in the following parts.

1. *“First of all, the idea of integrating two functional materials to develop multi-responsive materials is not new. For example, in a paper recently published in Advanced Functional Materials (Adv. Funct. Mater. 2017, 27, 1702784), the authors also combined ferrocene derivative with a thermoresponsive polymer to develop a thermal/redox dual-functional material for smart windows. Depending on the salt concentration, the material becomes opaque at a temperature ranging from 33-66 °C, demonstrating a better temperature tunability than the material reported in this manuscript.”*

Response:

Thanks a lot for the profound comments. The paper recently published in *Advanced Functional Materials* (*Adv. Funct. Mater.* **2017**, *27*, 1702784) reported a very nice work. In their work, a thermal/redox dual-functional material for smart windows was successfully fabricated based on organometallic poly(ionic liquid)s. The switching of light transmittance could be achieved by temperature regulation or redox stimuli. That's a very delicate design. However, redox stimuli in our work were used to switch between warm color and cool color, which could provide a more comfortable in-door environment to improve the feelings and emotions of indoor people. Meanwhile, both in warm and cool tones, temperature regulation could be achieved by switching the transmittance of **Fc-gel•EGP6** hydrogel between transparency and opacity. Notably, these two functions were cooperative and non-interfering.

Moreover, although ferrocene was used both in their paper and our work, the designs and applications of ferrocene were quite different. In their paper, ferrocene moiety was used to synthesize organometallic poly(ionic liquid)s and change the LCST-type transition temperature via redox stimuli, which could achieve the switching of light transmittance. However, in our work, the ferrocene group was not only applied to construct **EGP6**-ferrocene supramolecular inclusion complexes based on host–guest interaction to achieve the dramatically enhanced swelling and transparency of the hydrogel material, but also was used to switch the color of the material between warm and cool tune to provide a more comfortable in-door environment for improving the feelings and emotions of indoor people.

Furthermore, an orthogonal integration strategy of different properties was proposed for the first time. In our work, the thermo-responsiveness of **EGP6** and the redox-induced reversible color switching between warm and cool tune of ferrocene/ferrocenium units are orthogonally integrated into a supramolecular system to achieve the fabrication of warm/cool tone-switchable thermochromic materials with cooperative and non-interfering dual functions. So we believe that this orthogonal integration strategy of different properties could provide a new direction for design and development of novel functional materials.

2. *“Second, for smart window applications, the change of color by adding/immersing into redox species is not quite feasible. Also, the color change time (8 minutes) seems too long for real life applications. If the material can be electrochromic and can change its color at a faster speed, the importance of this work would be largely enhanced.”*

Response:

Many thanks for the profound comments and helpful suggestions. Just as Reviewer 2 said, a feasible and practical way for smart window is very important for the real-life applications, such as electrochemical triggers. In this **Fc-gel•EGP6** hydrogel system, thermochromic performance depends on the environmental

temperature, while color switching between warm and cool tune mostly depends on the human's comfort demands and controlled by a feasible redox stimuli. According to the previous work (*Chem. Commun.*, **2013**, 49, 5085-5087; *Macromolecules* **2015**, 48, 4403-4409; *J. Am. Chem. Soc.* **2010**, 132, 9268-9270), electrochemistry methods including cyclic voltammetry (CV) and square-wave voltammetry (SWV) studies indicated that the pillararene-ferrocene inclusion complexes have stable oxidation and reduction peaks. Therefore, we have fabricated an electrochemical device based on **Fc-gel•EGP6** hydrogel to achieve the color switching between warm and cool tune by electrochemical triggers. As shown in Figure R4, after being immersed in 0.1 mol/L KCl aqueous solution, the **Fc-gel•EGP6** hydrogel was sandwiched between two pieces of ITO conductive glass to form a simple device. By holding the potential at +4.0 V for 6 min, the color of the material in the device could be completely transformed from orange to green due to the electrochemical oxidation of ferrocene to ferricenium cation. After further holding the potential at -4.0 V for 6 min, **Fc-gel•EGP6** hydrogel was reduced and the color of the material returned to orange again. The above investigations have been added in the revised Manuscript as Fig. 7.

Figure R4. (a) Schematic illustration of the electrochemical device based on **Fc-gel•EGP6** hydrogel; (b) Photographs of the device in reduced and oxidized states, respectively.

3. “Lastly, although the authors showed some reproducibility result, the number of cycles is not enough for applications like smart windows. Reproducibility experiments with more cycles are encouraged.”

Response:

Thanks a lot for the profound suggestions. Actually, we have investigated much more cycles, the original repeatability experiments were only the first ten cycles. According to Reviewer 2’ suggestion, we have provided the whole cyclic experiments in the revised Manuscript as Fig. 6. The warm/cool tone-switchable function and optical performance of **Fc-gel•EGP6** hydrogel were hardly altered after more than 100 thermal or redox cycles as shown in Figure R5. Thus, the **Fc-gel•EGP6** hydrogel exhibited high reproducibility and good reversibility in the processes of warm/cool tone-switching and thermochromism..

Figure R5. Transmittance changes of (a) **Fc-gel•EGP6** hydrogel and (b) oxidized **Fc-gel•EGP6** hydrogel at 1099 nm by alternately exposing to water at 25 °C and 40 °C. (c) Color changes and (d) transmittance changes at 1099 nm and 620 nm of **Fc-gel•EGP6** hydrogel by alternately exposing to $(\text{NH}_4)_2\text{S}_2\text{O}_8$ (cool color) and $\text{N}_2\text{H}_4\cdot\text{H}_2\text{O}$ (warm color) aqueous solution at 25 °C.

Reviewer 3

Comments to the Author

“Manuscript by Sai Wang et al., “Warm/cool tone switchable thermochromic material for smart windows by orthogonally integrating the unique properties of pillar[6]arene and ferrocene ” presents a novel orthogonal

integration strategy of different properties for smart windows. The thermo-responsiveness of ethylene glycol chain-modified pillar[6]arene (EGP6) and the redox-induced reversible color switching between warm and cool tone of ferrocene/ferrocenium groups are integrated into a system to achieve the fabrication of warm/cool tone-switchable thermochromic materials with cooperative and non-interfering dual functions.

However, some details are not sufficient for publishing in the present form and major revision are needed. The following questions should be addressed before publishing.”

Response:

We greatly appreciate Reviewer 3 for his/her positive evaluation and kind recommendation on publishing our manuscript in *Nature Communications* after major revisions. And all the concerns of Reviewer 3 have been addressed in detail as follows, and the corresponding revisions have also been made in the revised Manuscript and Supplementary Information.

General issues:

1. *“The UV-Vis-NMR measurement is not in the full spectrum of solar energy (to 2500 nm). Energy modulation need to be considered from visible range to NIR range.”*

Response:

Many thanks for the profound suggestions. Transmittance spectra of **Fc-gel-EGP6** hydrogel with the wavelength range from 300 to 2000 nm were measured at 25 °C and 40 °C, which were provided in the revised Supplementary Information as Supplementary Fig. 14. The result also revealed that the **Fc-gel-EGP6** hydrogel had an excellent property of thermochromism. However, due to the limitation of our current instrument, the transmittance spectra could only be measured from 300 to 2000 nm, but it had almost no influence on our results. In addition, we found that some reported work (*Adv. Funct. Mater.* **2017**, *27*, 1702784; *Adv. Mater.* **2010**, *22*, 468–472; *Adv. Energy Mater.* **2017**, *7*, 1602209; *Sci. Rep.* **2015**, *5*, 11773; *Sci. Rep.* **2015**, *5*, 7646; *J. Mater. Chem. A*, **2016**, *4*, 6064-6069) also used partial UV-Vis spectra to investigate the transmittance of the obtained materials.

Figure R6. Transmittance spectra of **Fc-gel•EGP6** hydrogel at 25 °C and 40 °C, respectively.

2. “ From the paper, researchers only mentioned the thickness of dry *Fc-gel*, can you provide the thickness of samples in Fig 4? The very low luminous transmission is a concern. Reducing the thickness should help to enhance the luminous transmission to be used in thermochromic materials.”

Response:

Many thanks for the helpful suggestions. The thickness of **Fc-gel** immersed in pure water is about 1 mm, and the thickness of the formed **Fc-gel•EGP6** hydrogel in Fig. 4 is about 2.8 mm. The above data were also provided in the Figure Caption of Fig. 4 in the revised Manuscript. Moreover, we will adopt Reviewer 3’s nice suggestion to further reduce the thickness of the hydrogel, and even try to get a membrane-like hydrogel material in the future study. Hopefully, this will greatly improve the luminous transmission of the obtained material for its real applications as thermochromic materials.

3. “Durability is always a problem for organic smart window, for the durability test, 10 cycles were not enough for both transmittance change and redox reaction, is it possible to test at least 50 cycles or its better to test until the gel will not change color and transmittance to understand the limitation of this material (Fig 6)”

Response:

Many thanks for the profound suggestions. Just as aforementioned (Figure R5), we have investigated the durability of **Fc-gel•EGP6** hydrogel for more than 100 thermal or redox cycles. The results showed that **Fc-gel•EGP6** hydrogel exhibited high reproducibility and good reversibility in the processes of transmittance change and warm/cool tone-switching, which were provided in the revised Manuscript as Fig. 6.

4. “Since it is a new type of thermochromic material, please characterize it in a standard way, please calculate the T_{lum} and ΔT_{sol} . and the calculation details is listed in the paper <https://doi.org/10.1016/j.jallcom.2017.10.045> and compare with other thermochromic materials as summerized in [10.1039/C6TC02694J](https://doi.org/10.1039/C6TC02694J).”

Response:

Thanks a lot for the profound suggestions and useful references. According to the helpful suggestion of Reviewer 3, the solar modulation ability (ΔT_{sol}) and enhanced luminous transmittance (T_{lum}) of **Fc-gel•EGP6** hydrogel were calculated in the revised Supplementary Information (Part S9) according to the reference method (*J. Alloy. Compd.* **2018**, 731, 1197-1207). Compared with the other reported results for the mostly studied inorganic VO_2 thermochromic materials (*J. Mater. Chem. C*, **2016**, 4, 8385-8391), an unprecedented good combination of higher T_{lum} (64.0%) and ΔT_{sol} (66.9%) was obtained for this hydrogel material. Moreover, the above descriptions and corresponding references were also added in the revised Manuscript as Ref. 54 and 55.

5. “The authors provided a new idea to integrate the thermochromic composite with the color switchable compound. However, the idea is lack of proofing-of-concept. The authors demonstrated the gels with different oxidation state by soaking into different precursors. However, the capability of such a color-switch approaching to the real-life application has not been investigated. I suggest that the authors could fabricate device to show a demo that based on this new material to proof the concept”

Response:

Many thanks for the profound comments and helpful suggestions. Considering the real-life application and the electrochemical properties of ferrocene moiety (*Chem. Commun.*, **2013**, 49, 5085-5087; *Macromolecules* **2015**, 48, 4403-4409; *J. Am. Chem Soc.* **2010**, 132, 9268-9270), we have fabricated an electrochemical device based on **Fc-gel•EGP6** hydrogel to achieve the color switching between warm and cool tune by electrochemical triggers. Just as aforementioned (Figure R4), after holding the potential at +4.0 V for 6 min, the color of the material in the device could be completely transformed from orange to green due to the electrochemical oxidation of ferrocene to Fc^+ . Meanwhile, by further holding the potential at -4.0 V for 6 min, **Fc-gel•EGP6** hydrogel was reduced and the color of the material returned to orange again. And the above investigations have also been added into the revised Manuscript as Fig. 7.

6. “To continue from previous point, How to immerse the hydrogel film into the respective solution for redox reactions in real application? Whereas, since this is a research paper, these concerns are less critical. Maybe share a few strategies to put this material in real applications.”

Response:

Many thanks for the profound comments. A feasible and practical way such as electrochemical triggers for fabricating smart window is very important for putting the material in real applications. For this supramolecular system, there are two strategies that may put this hydrogel material in real applications. The first one is to prepare a thin film as the coating material for smart windows, then the redox medium could be sprayed on the surface of the film, which would be feasible to achieve the color switching of the windows. Another strategy is the electrochemical method. Just as we have mentioned above, a simple electrochemical device based on **Fc-gel•EGP6** hydrogel was fabricated to achieve the color switching between warm and cool tune by electrochemical triggers, indicating that this hydrogel material may hold great potential for fabricating smart windows in real life.

Once again, many thanks to the editors and referees for the profound comments and suggestions on improving the quality of our manuscript, and we are very grateful for your time and your continuous effort in the processing of our manuscript.

Yours sincerely,

Leyong Wang

Reviewers' Comments:

Reviewer #1:

Remarks to the Author:

The revised version of this very nice manuscript, with high scientific value and novelty, can be published as it is now. All comments raised by the reviewers have been well addressed.

Reviewer #2:

Remarks to the Author:

The added electrochromic experiment suggested that this system has the potential to be applied in smart windows application. Because this work represents one of the first examples of utilizing supramolecular materials with high modularity for smart windows applications, it is suggested that the authors could include more references (especially latest work) about thermochromic supramolecular polymers and smart windows (Nat. Mater. 2018, 17, 261; Chem. Mater. 2017, 29, 9937-9945; Adv. Funct. Mater. 2017, 27, 1702784; Adv. Funct. Mater. 2016, 26, 8604-8612). I would recommend acceptance of this manuscript in Nature Communication after a more comprehensive introduction, especially regarding the recent development of supramolecular thermochromic materials, is included.

Reviewer #3:

Remarks to the Author:

All my questions have been addressed well. ITs a good piece of work.

The following is a point-to-point response to the reviewers' comments (for your convenience, we repeat the referee's comments below in *black*, followed by our reply in blue).

Referee 1

Comments to the Author

"The revised version of this very nice manuscript, with high scientific value and novelty, can be published as it is now. All comments raised by the reviewers have been well addressed."

Response:

We greatly appreciate Referee 1 for the positive evaluation and kind recommendation of publishing our manuscript in *Nature Communications*.

Referee 2

Comments to the Author

“The added electrochromic experiment suggested that this system has the potential to be applied in smart windows application. Because this work represents one of the first examples of utilizing supramolecular materials with high modularity for smart windows applications, it is suggested that the authors could include more references (especially latest work) about thermochromic supramolecular polymers and smart windows (Nat. Mater. 2018, 17, 261; Chem. Mater. 2017, 29 9937-9945; Adv. Funct. Mater. 2017, 27, 1702784; Adv. Funct. Mater. 2016, 26, 8604-8612). I would recommend acceptance of this manuscript in Nature Communication after a more comprehensive introduction, especially regarding the recent development of supramolecular thermochromic materials, is included.”

Response:

We greatly appreciate Referee 2 for the positive comments and kind recommendation of publishing our manuscript in *Nature Communications* after minor revisions, and many thanks for the profound suggestions of Referee 2. As suggested, these excellent work related to thermochromic materials (*Nat. Mater.* **2018**, 17, 261), thermochromic organic materials (*Adv. Funct. Mater.* **2017**, 27, 1702784), and color switchable supramolecular materials (*Adv. Funct. Mater.* **2016**, 26, 8604-8612; *Chem. Mater.* **2017**, 29, 9937-9945) has been cited as ref. 16, ref. 30, ref. 33, and ref. 34 in the revised Manuscript, respectively. Moreover, we have made a more comprehensive introduction regarding the recent development of thermochromic supramolecular materials based on charge-transfer interactions in the Introduction part of the revised Manuscript.

Referee 3

Comments to the Author

“All my questions have been addressed well. ITs a good piece of work.”

Response:

We greatly appreciate Referee 3 for the positive comments and kind recommendation of publishing our manuscript in *Nature Communications*.